# A Machine Learning Framework to Predict the Tensile Stress of Natural Rubber: Based on Molecular Dynamics Simulation Data

**DOI:** 10.3390/polym14091897

**Published:** 2022-05-06

**Authors:** Yongdi Huang, Qionghai Chen, Zhiyu Zhang, Ke Gao, Anwen Hu, Yining Dong, Jun Liu, Lihong Cui

**Affiliations:** 1College of Mathematics and Physics, Beijing University of Chemical Technology, Beijing 100029, China; 2020201070@buct.edu.cn (Y.H.); 2021201051@buct.edu.cn (A.H.); 2State Key Laboratory of Organic-Inorganic Composites, Beijing University of Chemical Technology, Beijing 100029, China; 2020200508@buct.edu.cn (Q.C.); zhangzy59908@163.com (Z.Z.); 18242361087@163.com (K.G.); 3School of Data Science and Hong Kong Institute for Data Science, Centre for Systems Informatics Engineering, City University of Hong Kong, Tat Chee Avenue, Kowloon, Hong Kong

**Keywords:** natural rubber, tensile stress, XGBoost, molecular dynamics simulation, nearest-neighbor interpolation, SMOTE

## Abstract

Natural rubber (NR), with its excellent mechanical properties, has been attracting considerable scientific and technological attention. Through molecular dynamics (MD) simulations, the effects of key structural factors on tensile stress at the molecular level can be examined. However, this high-precision method is computationally inefficient and time-consuming, which limits its application. The combination of machine learning and MD is one of the most promising directions to speed up simulations and ensure the accuracy of results. In this work, a surrogate machine learning method trained with MD data is developed to predict not only the tensile stress of NR but also other mechanical behaviors. We propose a novel idea based on feature processing by combining our previous experience in performing predictions of small samples. The proposed ML method consists of (i) an extreme gradient boosting (XGB) model to predict the tensile stress of NR, and (ii) a data augmentation algorithm based on nearest-neighbor interpolation (NNI) and the synthetic minority oversampling technique (SMOTE) to maximize the use of limited training data. Among the data enhancement algorithms that we design, the NNI algorithm finally achieves the effect of approaching the original data sample distribution by interpolating at the neighborhood of the original sample, and the SMOTE algorithm is used to solve the problem of sample imbalance by interpolating at the clustering boundaries of minority samples. The augmented samples are used to establish the XGB prediction model. Finally, the robustness of the proposed models and their predictive ability are guaranteed by high performance values, which indicate that the obtained regression models have good internal and external predictive capacities.

## 1. Introduction

It is well known that natural rubber (NR) is an essential biopolymer with unique properties, including excellent elasticity and effective heat dispersion, abrasion, impact resistance and resilience [1,2,3,4,5]. Notably, due to its outstanding properties, NR cannot be replaced by synthetic rubber, especially in many fields, such as military and medical devices [6].

The excellent properties of vulcanized NR are closely related to its special structures. To clarify the structure–mechanics relationship, it is necessary to investigate the effects of the non-rubber components, especially proteins and phospholipids, on the mechanical properties of NR more precisely. The effects of phospholipids [7,8] and proteins [9,10] on the properties of NR have been reported. For instance, an analysis of rheological and stress relaxation spectra revealed that the hydrogen bonds between phospholipids and proteins at the chain ends reinforce the formation of the physical entanglement network of NR, while the reinforcement of this network is significantly weakened in NR with a low protein content [8]. In addition, it has also been shown that non-rubber components can promote the nano-dispersion of rubber particles and reinforce the tensile strength of vulcanized NR [11]. It is obvious that phospholipids and proteins have a great influence on mechanical properties, with the possible influencing factors being phospholipid and protein content (ω), and the strength of hydrogen bond interactions ($\varepsilon_{H}$) and the strength of non-hydrogen bond interactions ($\varepsilon_{NH}$) (strength of the interaction between phospholipids and proteins with the rubber matrix, respectively). Thus, adjusting the optimum configuration of ω-$\varepsilon_{H}$-$\varepsilon_{NH}$ is critical to improve the mechanical properties of NR.

However, although various types of advanced experimental characterization equipment and means are available, and many analyses have been conducted on the principles of the superior performance of NR, a systematical understanding of the structure–mechanics relationship of NR is still lacking. Furthermore, it is far from the goal of accurately and quantitatively establishing the relationship between the microstructure and macroscopic properties of NR, which seriously limits the performance optimization of elastomeric materials. In previous studies, it has been realized that molecular dynamics (MD) simulation is a powerful technique that can be used to provide insight into materials with complex compositions and reveal the structure–mechanics relationship at the molecular level [12,13,14,15,16]. Evidently, we have previously investigated the various effects of composites using MD simulations, such as filler morphology, interfaces between polymers and fillers, and polymer networks [17,18,19]. However, we have also found that in materials and chemistry science, the experimental data are usually concentrated in one or several particular regions of the data space because of the constraints of actual manufacturing and experimental design. This means that the dataset is generally not large or is even almost without high-quality data points [20]. This is also the case for MD simulations, which seem to be a general problem in most property prediction problems. The traditional methods for polymer material research often confront both problems, not only involving expensive and lengthy experimental periods but also requiring complex and laborious calculations. Fortunately, with the rapid development of machine learning (ML) theory and applications, increasingly more ML techniques have been applied in the polymer material field, especially after a great project called the “Materials Genome Initiative” was proposed in 2011 [21]. ML models have been widely applied to predict molecular properties and develop force fields for MD simulations [22,23,24,25], and they are beginning to receive attention in the design of bio-inspired composites. For example, Chen analyzed the results of MD simulations using ML and developed a new density-based trajectory clustering (DTC) method to elucidate the Li diffusion mechanism within a Li_7_La_3_Zr_2_O_12_ (LLZO) crystal lattice [26]. Lu combined ML techniques and density flooding theory calculations in order to develop a target-driven approach to predict undiscovered hybrid organic–inorganic perovskites (HOIPs) in photovoltaics [27]. Moreover, A. Rahman developed a novel convolutional neural network (CNN)-based framework to predict the shear strength of carbon nanotube–polymer interfaces based on molecular dynamics simulation data [28].

From the above discussion, it is clear that MD simulation-based mechanical performance prediction for NR requires a large number of high-quality data points. Herein, for the first time, we employ a molecular dynamics simulation to examine the effects of the key structural factors on the tensile stress of NR at the molecular level. We use coarse-grained molecular dynamics (CGMD) simulations to study the stress–strain behavior of NR with a focus on three factors: phospholipid and protein total mass fraction (ω), the strength of the hydrogen bond interaction ($\varepsilon_{H}$) and the strength of the non-hydrogen bond interaction ($\varepsilon_{NH}$). Since molecular dynamics simulations of natural rubber are computationally too time consuming and it is difficult to fully examine the effects of various factors on mechanical properties, an alternative method is needed to accurately and efficiently simulate molecular dynamics simulations. In this work, based on the data we obtain by simulating NR via MD simulations, we propose a brand-new framework combined with MD simulations, a data augmentation algorithm based on nearest-neighbor interpolation (NNI), a synthetic minority oversampling technique (SMOTE) and an extreme gradient boosting (XGB) model to predict the tensile stress of NR. The objectives of the framework we design are (i) to accelerate the prediction of the tensile stress of NR; (ii) to analyze the importance index of the ω, $\varepsilon_{H}$ and $\varepsilon_{NH}$ variables in the system with respect to the tensile stress of NR; and (iii) to use the methods and results of this this study as benchmarks for subsequent studies focusing on ML models to predict other mechanical properties of NR based on MD data.

As with the majority of problems encountered in the materials and chemical sciences, we encounter a situation of insufficient experimental data for MD. In general, with very small training samples (hundreds, tens or even a few samples), existing machine learning and deep learning models are commonly unable to achieve complex model building through iteration, and models trained with small samples can easily fall into overfitting small samples and underfitting the target task, resulting in poor overall model performance. Models built in such a form may even yield completely wrong results. Currently, there are three mainstream approaches that can be used to solve the small sample problem: data augmentation (Data), reducing the space that the model needs to search (Model) and optimizing the process of searching for the optimal model (Algorithm) [29]. We choose data augmentation, which is a relatively intuitive and effective method, to solve the problem of small sample size. NNI and SMOTE, which are traditional data enhancement algorithms originally applied to image processing, are employed in our data enhancement algorithms. They are used to solve the problems of sample insufficiency and imbalance, respectively. After data enhancement, the original samples are expanded from 86 to 2188. Meanwhile, the data points are more evenly distributed in the sample space, which is more favorable for XGB to learn information and predict target values, and the enhanced samples are used to build XGB prediction models. As a comparison, we establish multiple linear regression (MLR) and Support Vector Regression (SVR) models with the same enhanced data to compare the prediction results and model performance of each model. After obtaining the tensile stress prediction results, we also use the SHapley Additive exPlanation (SHAP) method to conduct feature importance analysis, and the results show that $\varepsilon_{H}$ contributes the most to tensile stress, followed by ω and $\varepsilon_{NH}$. Finally, the robustness of the proposed models and their predictive ability are guaranteed by high performance values. These values indicate that the obtained regression models have good internal and external predictive capacities.

## 2. Molecular Modeling

MD simulation is an accepted computational method, and we apply it in this research to establish an NR coarse-grained model aimed at generating the raw input data for XGB. For the proposed method, MD is applied to simulate the tensile stress of NR.

### 2.1. NR Modeling

We establish the NR coarse-grained model, as shown in Figure 1. The NR molecular chains are represented by the bead-spring model, in which the monomers are simplified into spherical beads and the beads are connected through harmonic potential. The NR single chain consists of three kinds of beads, namely, a phosphate group, cis-1,4-polyisoprene repeating unit and a polypeptide. Furthermore, to simulate the supramolecular network inside the NR, we introduce phospholipids and proteins into the system, where hydrogen bonds form between phospholipids or proteins and the chain ends (phosphate groups and polypeptides), as shown in Figure 1a. At the same time, hydrogen bonds also form among the phospholipids or proteins themselves. The supramolecular network can be finally constructed due to the hydrogen bonds (Figure 1b), which help form two kinds of clusters serving as temporary cross-linked sites (Figure 1c). In our systems, the number of phospholipid and protein beads are set to be identical, and when the number of phospholipid and protein beads is 0, it corresponds to a PIP system.

To simplify the coarse-grained NR model, the mass of all five kinds of beads mentioned above is identical. The single chain consists of 331 beads, whose molar mass is 22,508.0 g⋅mol^−1^. The NR system contains 100 identical single chains, and the number of free phospholipids and proteins is a variable. In addition, we also establish a system consisting of only 100 identical PIP chains, whose molar mass is 22,508.0 g⋅mol^−1^, as a comparison [30].

We investigate the mechanical properties of the cross-linked systems. To obtain cross-linked NR and PIP networks, extra cross-linked bonds are introduced to the above systems. The cross-linking in our model refers to the formation of the covalent bonds between adjacent coarse-grained beads from different chains. Specifically, the cross-linked network is formed by the random selection of two beads with a distance of less than 5.5 Å. Since it is not our aim to study a specific cross-linking or vulcanization process, we merely build the cross-linked network and do not invoke a probabilistic approach to introduce the cross-linked bonds [30,31]. To keep the cross-linked density identical in every system, 468 covalent bonds, regarded as 2 phr sulfur in the form of a trisulfide bond, are introduced to every system (100 phr).

In our coarse-grained simulations, the non-bonded interaction between polymer beads is described by the LJ 12-6 potential [32]. Meanwhile, to simplify the system, the hydrogen bond interaction is also modeled as follows:(1)$$U_{ij}\left(r\right)=4\varepsilon \left[{\left(\frac{\sigma_{0}}{r}\right)}^{12}-{\left(\frac{\sigma_{0}}{r}\right)}^{6}\right],\;\;r<r_{c}$$
where ε is the depth of the potential well, $\sigma_{0\;}$ is the finite distance at which the inter-particle potential is zero, $r_{c}$ is the cutoff distance, and r is the distance between the beads for the LJ potential. Among the pair interactions, two factors, namely, (i) the strength of the hydrogen bond interaction ($\varepsilon_{H}$) and (ii) the strength of the pair interaction of cis-1,4-polyisoprene repeating unit with phospholipids and proteins (non-hydrogen bond interaction, $\varepsilon_{NH}$), are investigated. All the other non-bonded interaction strength values are set to 0.38 kcal·mol^−1^, except for the two interaction strength factors mentioned above. For all systems, the over-zero distance of the potential is 4.89 Å and the cutoff distance is 12.5 Å.^30^. The detailed corresponding force field parameters of the hydrogen bond and non-bonded interaction are shown in Table 1 and Table 2.

The bond stretching energy between the adjacent beads is modeled via the harmonic potential:(2)$$U_{stretch}\left(l\right)=k_{stretch}{\left(l-l_{0}\right)}^{2}$$
where $k_{stretch}$ is the stiff constant for bond stretching, l is the current bond length, and $l_{0}$ is the equilibrium bond length. To simplify the simulation, $k_{stretch}$ and $l_{0}$ in all types of covalent bonds are set to 3.2 kcal·mol^−1^ and 4.6 Å, respectively [30].

The angle potential in all our systems is fitted with the harmonic potential:(3)$$U_{bend}\left(\theta \right)=k_{bend}{\left(\theta -\theta_{0}\right)}^{2}$$
where $k_{bend}$ is the spring constant for angular motion, θ is the current angle, and $\theta_{0}$ is the equilibrium angle. All angle potentials in all our systems are set to $k_{bend}$ = 1.53 kcal·mol^−1^, $\theta_{0}$ = 121° [30].

### 2.2. Molecular Dynamics Simulation Methods

Our CGMD simulations start from a nonoverlapping configuration of all beads in a large simulation box. Periodic boundary conditions in three directions are also employed in the simulations. Then, we use an NPT ensemble to equilibrate all systems by adjusting the Nosé–Hoover temperature thermostat and pressure barostat. The temperature and pressure in the simulations are set to 298 K and 0.1 MPa, respectively. The velocity-Verlet algorithm is used to integrate the equations of motion to describe the motion of all beads with the time unit $\delta^{*}$ = 1 fs. The obtained structure is further equilibrated under the NPT ensemble to ensure that each molecular chain moves at least $2R_{g}$ ($R_{g}$ is the root-mean-squared radius of gyration of the polymer). Finally, the size of the systems becomes gradually steady. After that, the systems are annealed at 400 K to relax the conformation for 10 ns and further equilibrated in the NPT ensemble to collect trajectory and essential data.

After equilibration, uniaxial tensile deformation is performed to obtain stress–strain ($\sigma_{T}$-$\varepsilon_{T}$) curves in the CGMD simulations, and it was also utilized in our previous studies [30,31]. The simulation box is stretched in the Z-direction at a constant engineering strain rate ${\dot{\varepsilon }}_{T}$ (Equation (4)), while the lengths in the X-directions and Y-directions are decreased to keep the volume of the simulation box constant.
(4)$${\dot{\varepsilon }}_{T}=\left(L_{Z}\left(t\right)-L_{Z}\left(0\right)\right)/\left(L_{Z}\left(0\right)\cdot t\right)=\varepsilon_{T}/t$$
where $\;L_{Z}\left(t\right)\;$ and $L_{Z}\left(0\right)$ are the box length in the Z-direction at time t and the beginning, respectively, and $\varepsilon_{T}$ is the tensile strain. The strain rate is set to 10^−9^/s [32]. The tensile stress is expressed through the bias tensor of the pressure as [33]
(5)$$\sigma_{T}=\left(1+\nu \right)\left(-P_{\mathrm{ZZ}}+\sum_{i}P_{\mathrm{ii}}/3\right)\approx \left(-3P_{\mathrm{ZZ}}+\sum_{i}P_{\mathrm{ii}}\right)∕2$$
where $\sum_{i}P_{\mathrm{ii}}/3$ represents the isostatic pressure of the system and ν represents Poisson’s ratio of the system.

All CGMD simulations are carried out using the large-scale atomic/molecular massively parallel simulator (LAMMPS), developed by Sandia [34].

## 3. Proposed Machine Learning Framework

Each MD simulation of NR tensile stress requires a significant amount of computational effort and time (approximately 300–1500 core hours). Therefore, we aim to accelerate the predictions of tensile stress by developing a novel XGB-based framework. We describe our proposed method in detail in this section.

### 3.1. MD Experimental Data Collection

As an elastic material, NR is widely used in industry. Tensile capacity is one of the most important properties of NR, and it is of great significance to study the tensile capacity of NR composites. In this work, we obtain 86 sets of data for NR using the method in Section 2, where ω, $\varepsilon_{H}$ and $\varepsilon_{NH}$ are the feature values, and tensile stress is the label value. In this experiment, we select 600% tensile stress.

### 3.2. Data Preprocessing and Feature Engineering

One of the challenges of the problem presented in this research is the limited size of data arising due to the high computational cost associated with MD simulations. Therefore, we need to design a method to extract high-value information from the limited training data. In this work, we leverage the NNI-SMOTE algorithm to augment our original dataset. The algorithm is a basic structure, and it can be divided into two steps. In the first step, the dataset is enhanced with the NNI algorithm. In the second step, after data enhancement, the imbalance samples are processed with the SMOTE algorithm. NNI and SMOTE are the traditional algorithms used for data enhancement, and they were originally applied in image processing [35,36]. The former is a k-dimensional tree (KD-tree)-based interpolation algorithm, and the latter is an oversampling technique proposed by Chawla in 2002 [37]. In this study, they are used to solve the problems of sample insufficiency and imbalance, respectively.

#### 3.2.1. Data Expansion

Before proceeding NNI, we first construct an interpolation grid based on the original samples and then achieve the effect of approaching the original data sample distribution by interpolating at the neighborhood of the original sample, aiming to fully obtain the information in a small neighborhood of each sample point. We assume that all interpolation points obey the normal distribution $N\left(\mu ,\sigma^{2}\right)$ when we interpolate. The mean and variance values are described as
(6)$$\mu =x_{ij}$$
(7)$$\sigma^{2}=\frac{\lambda }{m}\Sigma_{i=1}^{m}x_{ij}$$
where $x_{ij}$ is the value of the j-th feature of the i-th sample, m  is the number of samples, and λ is an interpolation parameter that can change the size of the interpolated area and is positively correlated with the interpolated area. Choosing a suitable λ value is essential for the following work. After several experimental comparisons, we select the optimal interpolation parameters; the value of λ is set to 0.1, and 20 points are interpolated for each sample.

The chosen interpolation parameter λ is neither too large nor too small. Obviously, as the interpolated area continuously increases with λ, although more information is obtained, this also means that more noise is introduced. However, when the interpolation parameter λ is small, the newly interpolated points overlap on the original sample, meaning that we cannot fully obtain the information in the small neighborhood of each sample point. In order to analyze the interpolation results properly, we use principal component analysis (PCA) to reduce the dimensionality of the new dataset and to plot the sample distribution. Figure 2 presents a comparison of the data distribution after choosing a different interpolation parameter λ. The total number of samples in the dataset increases from 86 to 1806 after NNI.

#### 3.2.2. Solving Sample Imbalance

In practice, due to the constraints of actual manufacturing and experimental design, experimental data are usually concentrated in one or several particular regions of the data space. Therefore, during the experiment, no matter how much effort we put into making the dataset evenly distributed, the data distribution will still be unbalanced. This phenomenon is even more obvious after interpolation. It is worth noting that the sample imbalance reduces the learning ability of the model for clustered minority samples, because the learning target of ML prioritizes larger clustered samples. Hence, for better ML predictions, the Borderline-SMOTE algorithm is employed in this experiment to solve the problem of sample imbalance.

We divide the present work into two steps: First, we require a clustering analysis of our imbalanced dataset, and K-means is chosen as the clustering algorithm. Then, we use the SMOTE algorithm to interpolate at the clustering boundaries of minority samples. K-means has a critical parameter K (the number of clusters), and different K parameters will affect the effect of SMOTE interpolation. For the discussion of parameter K, please refer to Appendix A. After a series of experiments and comparisons, parameter K is set to 2. Therefore, the whole dataset is divided into two clusters, and the numbers of samples in the two clusters are 757 and 1049.

According to the results of the K-means algorithm, we use Borderline-SMOTE to further interpolate at the clustering boundaries of minority samples. For visualization of the results, we also use PCA to reduce the dimensionality and plot the distribution after SMOTE interpolation (Figure 3). The numbers of samples in the two clusters are 1007 and 1181 after Borderline-SMOTE interpolation.

### 3.3. Model Establishment and Improvement

In this work, an XGB model was established after adequate preprocessing of the original dataset. In order to ensure the predictive capability and stability of the XGB model, the 10-fold-cross-validation method was applied during the training/testing process. The 10-fold-cross-validation was performed as follows: (i) the whole dataset was randomly divided into 10 groups; (ii) then, 9 out of the 10 groups were randomly selected for training, and the remaining groups were used as test sets; (iii) in each iteration, the training-test was performed 10 times till each vector was used in the test set once; and, finally, (iv) the results of the 10 evaluations were averaged to reduce the error caused by the unreasonable selection of the test set.

For the XGB model to achieve the overall optimum value, we used the learning curve method to find the optimal parameters. In Figure 4, the different plots correspond to the different learning curves of the parameters. The abscissa axis represents different parameter values, and the ordinate axis represents the average $R^{2}$ of the 10-fold-cross-validation. $R^{2}$ is a value used to evaluate the performance of the model, and it is between 0 and 1; the closer the value is to 1, the better the model performance. Details about $R^{2}$ are described in Section 3.4. The final optimal parameter combination was the number of trees (“n tree”) = 55 (Figure 4a), the learning rates (“eta”) = 0.16 (Figure 4b), the maximum length from the root node to leaf node (“max depth”) = 8 (Figure 4c) and the L2 regularization parameters (“reg lambda”) = 120. All other parameters were selected as default values for the calculations.

### 3.4. Model Performance

As mentioned above, MD simulations can be expensive in computation. Therefore, the number of data points in the original dataset for this research is relatively small (86 data points). Hence, the performance of the MD-XGB model may vary depending upon the split of the training and test data. To investigate the variance in performance due to the methods of data segmentation, we evaluate 999 random instantiations of the train/test split. The metrics used to evaluate the performance of the established MD-XGB framework include the coverage (Cov) and coefficient of determination ($R^{2}$) [38].
(8)$$R^{2}=1-\frac{\sum_{i=1}^{N}{\left(X_{i}-{\hat{X}}_{i}\right)}^{2}}{\sum_{i=1}^{N}{\left(X_{i}-\bar{X}\right)}^{2}}$$
(9)$$\mathrm{Cov}=\frac{1}{N}\sum_{i=1}^{N}h\left[\frac{\left|X_{i}-{\hat{X}}_{i}\right|}{2\times sd\left(\left|X_{i}\right|\right)}\right],\;\;\;\;\;\;h\left(x\right)=\{\begin{array}{c} 1,\;\;\;\;\;\;\;\;\;x\leq 1\\ \;\;\;0,\;\;\;otherwise \end{array}$$
where ${\hat{X}}_{i}$ and $X_{i}$ are the predicted and the experimental values, respectively; $\bar{X}$ is the mean of experimental values; and $sd\left(\left|X_{i}\right|\right)$ is the standard deviation of the experimental values. Both $R^{2}$ and Cov are in the range of 0 and 1; larger values indicate that the model has a higher accuracy and robustness.

## 4. Results and Discussion

### 4.1. Performance Analysis of MD-XGB

With the performance metrics described in Section 3.4, we can now evaluate the predictive capability and stability of the established MD-XGB framework. The entire dataset is randomly divided into training and test sets 999 times. Figure 5 shows the regression plots for four of these instances (out of the 999 mentioned in Section 3.4). The plots show a comparison of the MD-XGB predicted results and the MD experimental data. If the regression line is nearer to the y = x line, the predicted value is closer to the true value, which means that the model performance is superior. We can see that the distribution between the XGB predictions and the MD experimental data remains generally consistent. The majority of the points are clustered near the regression line and are within the 95% confidence limit line. Only some scattered predictions largely deviate from the experimental values of MD, but the impact on the model is negligible.

The robustness of the proposed models and their predictive ability are guaranteed by the high $R^{2}$ based on bootstrapping, repeated 999 times, as shown Figure 6. The plots show that the $R^{2}$ values are found in the range of 0.94–0.98 and that the Cov values are 0.82–0.92. These values indicate that the obtained regression models have good internal and external predictive capacities.

XGB adds a regular term to the objective function to control the complexity of the model. It makes the learned model simpler and prevents overfitting. XGB has more accuracy and stability than other traditional machine learning algorithms. For proof of this, we perform two other experiments to compare the regression results of XGB with those of multiple linear regression (MLR) and Support Vector Regression (SVR), with all three models using the same training set and test set. Table 3 shows the Cov and $R^{2}$ of the three models. It can be clearly seen from Table 3 that XGB has the highest performance metrics, and this means that XGB has an absolute advantage in dealing with this type of problem.

### 4.2. Analysis of Variable Feature Importance

For the XGB model, we determine the importance of the features by the magnitude of the SHAP value of each variable. SHAP is a framework specifically designed to interpret model results, and it is widely used in machine learning, such as ensemble and deep learning models [39]. Figure 7a shows the ranking of feature importance. The features are ranked according to the average absolute value of SHAP. Among the three feature descriptors, $\varepsilon_{H}$ has the most significant effect on the tensile stress of NR, followed by ω and $\varepsilon_{NH}$. We can analyze this in more detail by examining Figure 7b, where a density scatter plot is drawn for all samples, and the x-axis represents different SHAP values. The wide areas indicate that there is a large number of aggregated samples. The colors on the right indicate the magnitude of the feature values, with red indicating high feature values and blue indicating low feature values. We can see that all variables show a positive correlation for the tensile stress of NR. Most of the sample points of $\varepsilon_{H}$ greatly affect the variation in tensile stress, because the SHAP value obviously increases or decreases with the value of $\varepsilon_{H}$. Additionally, ω is similar to $\varepsilon_{H}$, but ω has a portion of samples clustered around the 0 value of SHAP, so it has a much smaller effect on tensile stress than $\varepsilon_{H}$. For $\varepsilon_{NH}$, however, its effect on tensile stress is minimal because it has the vast majority of samples clustered around the 0 value of SHAP, and it can be seen that changes in $\varepsilon_{NH}$ barely affect tensile stress.

The mechanical reinforcement of bionic natural rubber relies mainly on orientation and strain-induced crystallization during stretching, which are strongly related to the adsorption of clusters to the chain ends [40]. Therefore, the mechanical reinforcement effect is mainly related to the strength (not easily pulled apart during stretching) and the number of clusters adsorbed to the chain ends.$\;\varepsilon_{NH}\;$ is the strength of the interaction of phospholipids and proteins with the rubber matrix and has almost no effect on the adsorption of clusters to the chain ends and, thus, on the orientation and crystallization of the molecular chains, so it has minimal effect on the mechanical properties. Both $\varepsilon_{H}$ and ω have a significant effect on the mechanical properties because increases in both $\varepsilon_{H}$ and ω favor the adsorption of clusters to the chain ends. $\varepsilon_{H}$ has the greatest influence on the information of clusters, because when $\varepsilon_{H}$ is small, clusters do not yet form and have little effect on the mechanical properties, while as $\varepsilon_{H}$ increases, clusters gradually form, the number of attractive chain ends increases, and the strength of the attraction becomes stronger. ω mainly affects the size of the clusters and does not affect the strength of cluster adsorption on the chain ends. As the cluster size increases, the number of clusters adsorbed on the chain ends increases, which plays a greater role in the mechanical properties. Therefore, we obtain a reasonable result indicating that $\varepsilon_{H}$ has the greatest effect on the mechanical properties, followed by ω and finally $\varepsilon_{NH}$.

### 4.3. Model Visualization and External Validation

According to results of feature importance in Section 4.2, we know that the value of $\varepsilon_{NH}$ has the least significant effect on the tensile stress of NR. This is also the case in practice, but we are more concerned with changes in the two variables $\varepsilon_{H}$ and ω and their effects on tensile stress (TS). Therefore, we first set the value of $\varepsilon_{NH}$ to any value between 0 and 1 (in the present research, we set $\varepsilon_{NH}\;$= 0.38), then predict 2.5 million combinations of $\varepsilon_{H}$-ω-TS using the established model and finally plot the 3-dimensional surface (Figure 8). We can see that there is an overall upward trend in the surface; this corresponds to the previous SHAP feature importance analysis. There are many other details that confirm the correct ranking of the importance of the features. For example, when $\varepsilon_{H}$ reaches a high value, a slight change in ω creates a huge change in TS. However, this is not the case when $\varepsilon_{H}$ reaches a low value.

We randomly select four intervals and simulate a new sample of 30 points as a validation set by MD within the interval. These points are used for external validation of the model (Figure 9a–d). The four panels in Figure 9 correspond to the four regions of A,B,C and D in Figure 8a. The plots show a comparison between the MD experimental values of the validation set and the predicted results. Most of the validation points are close to the predicted results; only some scattered predictions deviate from the predicted values. This also further verifies that previously mentioned: our models have good internal and external predictive capacities.

## 5. Conclusions

In this paper, we established an XGB model trained with MD data to predict the tensile stress of NR. Based on the MD dataset with the sample number of only 86 data points, the NNI algorithm was first used to enlarge the original data space. Then, K-means clustering analysis was performed on the expanded dataset, and the result showed that there was a problem of sample imbalance. In order to solve this problem, the SMOTE algorithm was applied to the dataset to perform a secondary interpolation, which further improved the distribution of data. Then, we established an XGB regression model consisting of tensile stress and three other factors ($\omega ,\;\varepsilon_{H}$ and $\varepsilon_{NH}$) of NR, and we analyzed the importance of the three factors on tensile stress using SHAP. Finally, the robustness of the establish model and its predictive capability were guaranteed by high performance values. These values indicate that the obtained regression models have excellent internal and external predictive capacities. We showed that the established framework can predict the tensile stress of NR based on MD data with reasonable accuracy.

The methods and the results of this this study can be used as benchmarks for subsequent studies focusing on ML models to predict other mechanical properties of NR based on MD data. The method is a framework, and the tensile stress is a label value in this experiment, and it can also be replaced by crystallinity or other mechanical properties. Our future work will focus on the development of ML methods that enhance the interpretability of ML models.

## Figures and Tables

**Figure 1 polymers-14-01897-f001:**
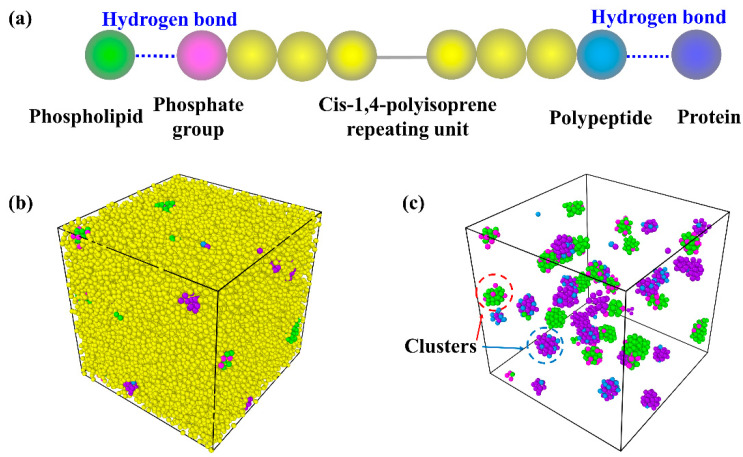
The coarse-grained model of NR. (**a**) The single chain, phospholipids and proteins of NR system; (**b**) the condensed state of NR; (**c**) the clusters formed by hydrogen bonds in the system.

**Figure 2 polymers-14-01897-f002:**
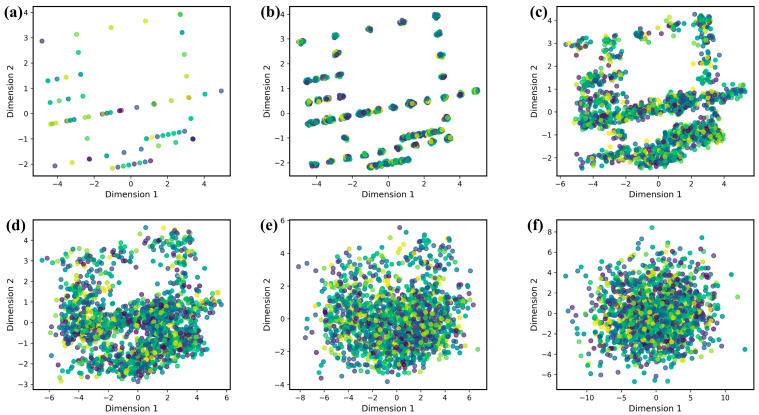
Comparison of data distribution plots after choosing different λ values for interpolation. (**a**) λ = 0, (**b**) λ = 0.05, (**c**) λ = 0.1, (**d**) λ = 0.2, (**e**) λ = 0.5, (**f**) λ = 1.

**Figure 3 polymers-14-01897-f003:**
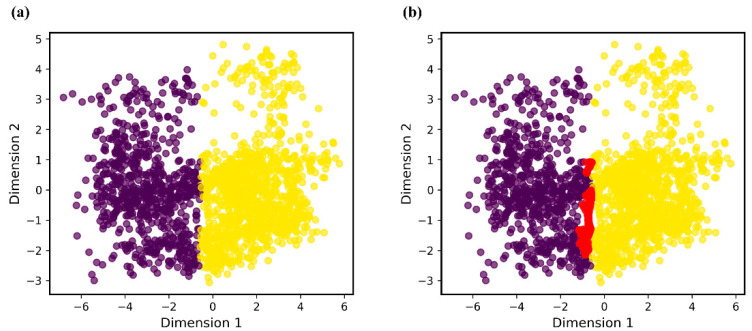
Comparison of (**a**) data distribution before SMOTE interpolation and (**b**) data distribution after SMOTE interpolation.

**Figure 4 polymers-14-01897-f004:**
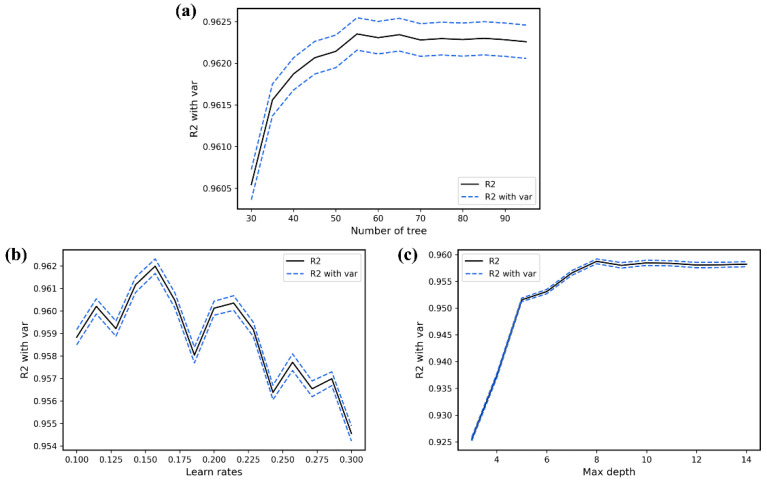
$R^{2}$ values for different parameter values in the improved XGB model. (**a**) nTree values from 1 iterate to 100, (**b**) learn rate from 0.1 iterates to 0.3, (**c**) max depth from 3 iterates to 15.

**Figure 5 polymers-14-01897-f005:**
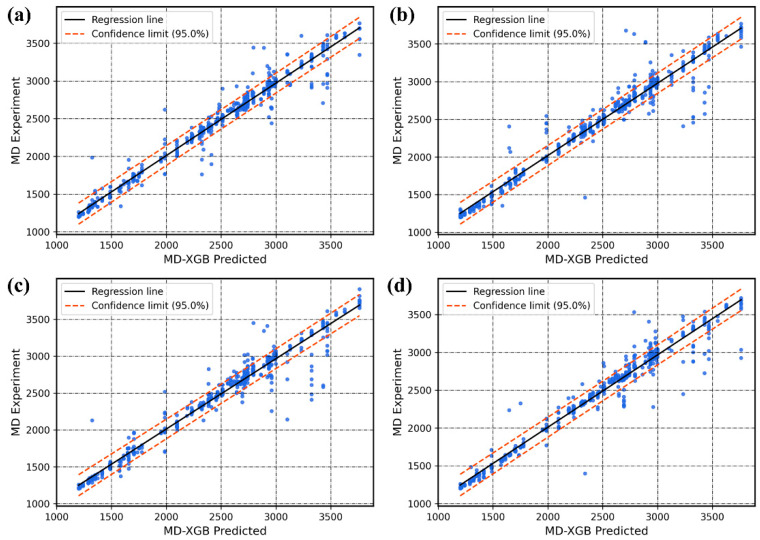
Sample regression plots comparing the MD experimental data with the MD-XGB predictions. Each plot in (**a**)–(**d**) corresponds to a different split of the training and test data. The $R^{2}$ and Cov values corresponding to each plot are (**a**) $R^{2}=0.960$, Cov = 0.916; (**b**) $R^{2}=0.962$, Cov = 0.863; (**c**) $R^{2}=0.962$, Cov = 0.898; (**d**) $R^{2}=0.968$, Cov = 0.897.

**Figure 6 polymers-14-01897-f006:**
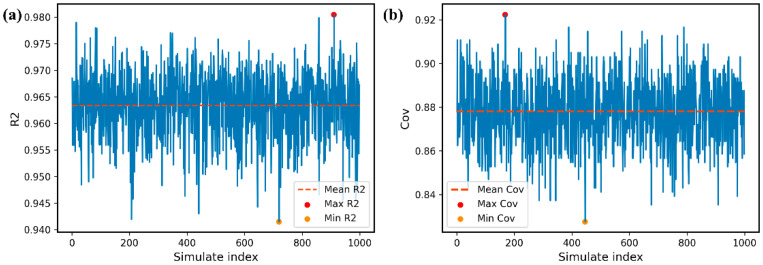
The evaluation of robustness using $R^{2}$ and Cov of predictive models. (**a**) The minimum, maximum and average values of $R^{2}$ are 0.941, 0.981 and 0.964, respectively; (**b**) The minimum, maximum and average values of Cov are 0.821, 0.922 and 0.878, respectively.

**Figure 7 polymers-14-01897-f007:**
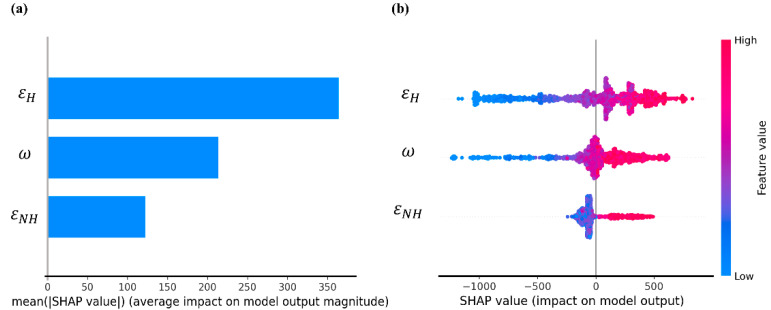
Feature importance analysis of each feature descriptor. (**a**) Feature importance is summarized with a bar chart, and the features are ordered by the average absolute value of SHAP; (**b**) feature importance is summarized with a density scatter plot.

**Figure 8 polymers-14-01897-f008:**
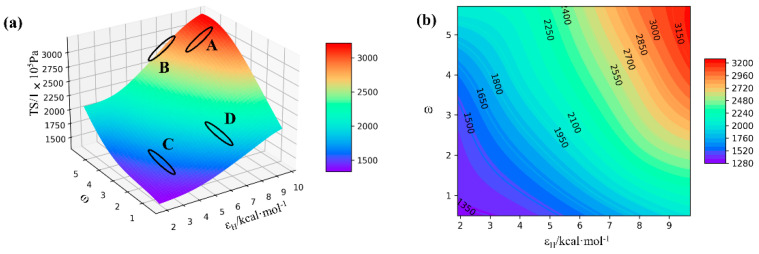
(**a**) MD-XGB predicted 3D image; (**b**) contour map of the predicted 3D image.

**Figure 9 polymers-14-01897-f009:**
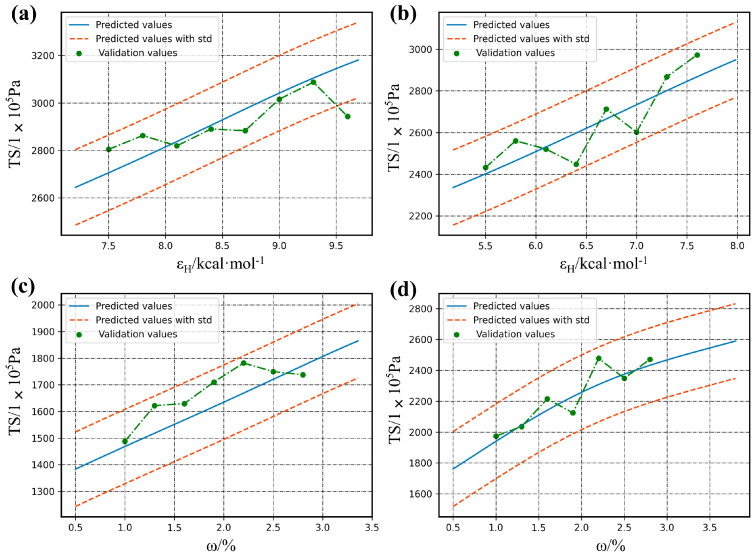
Comparison of the errors of the validation set and MD-XGB predictions, where (**a**–**d**) correspond to the four panels in Figure 8a.

**Table 1 polymers-14-01897-t001:** Potential parameters of the hydrogen bond interaction. The strengths of all hydrogen bonds are set to be identical, and the strength of the hydrogen bond is a variable in our study.

Hydrogen Bond Interaction				
Hydrogen Bond Pair		$\varepsilon_{H}/\mathrm{kcal}\cdot {\mathrm{mol}}^{-1}$	$\sigma_{0}/Å$	$r_{c}/Å$
Phosphate group	Phospholipid	3.80	4.89	12.5
Polypeptide	Protein	3.80	4.89	12.5
Phospholipid	Phospholipid	3.80	4.89	12.5
Protein	Protein	3.80	4.89	12.5

**Table 2 polymers-14-01897-t002:** Potential parameters of the non-hydrogen bond interaction. The strength of interaction between phospholipid or protein and cis-1,4-polyisoprene repeating unit (set to be identical) is a variable in our simulations.

Non-Hydrogen Bond Interaction				
Non-Hydrogen Bond Pair		$\varepsilon_{NH}/\mathrm{kcal}\cdot {\mathrm{mol}}^{-1}$	$\sigma_{0}/Å$	$r_{c}/Å$
cis-1,4-polyisoprene repeating unit	Protein	0.38	4.89	12.5
cis-1,4-polyisoprene repeating unit	Phospholipid	0.38	4.89	12.5

**Table 3 polymers-14-01897-t003:** Comparison of the $R^{2}$ and Cov of three types of models.

Metrics	Training Set			Testing Set		
	XGB	SVR	MLR	XGB	SVR	MLR
$R^{2}$	0.968	0.905	0.826	0.964	0.876	0.792
COV	0.886	0.828	0.800	0.878	0.835	0.787

## Data Availability

The raw/processed data required to reproduce these findings cannot be shared at this time, as the data also form part of an ongoing study.

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
