# Peer review of "A Machine Learning Framework to Predict the Tensile Stress of Natural Rubber: Based on Molecular Dynamics Simulation Data"

_polymers, 2022, doi:10.3390/polym14091897_

Round 1

Reviewer 1 Report

The manuscript describes a combination of computational methods for the derivation and expansion of tensile data for natural rubber (crosslinked polyisoprene (PI) chains, containing phosphate and polypeptide groups) in the presence of phospholipids and proteins. The original data come from coarse-grained (CG) molecular dynamics (MD) simulations and the goal is the determination of the optimum protein content that improves the mechanical properties. This is an interesting, well-written article both in terms of the used computational techniques and material for study. However, one of my concerns is related to the efficiency of the CG force field in the description of the thermodynamics of PI, e.g., density and bulk modulus compared both to the atomistic MD simulations and the experiment. This is crucial since these original MD data are augmented with interpolation techniques.

  • A comparison of the modulus with other MD simulations on PI is needed.
  • In 4.2 section it is mentioned that “The mechanical reinforcement of bionic natural rubber relies mainly on orientation and strain-induced crystallization during stretching, which are strongly related to the adsorption of clusters to the chain ends”. Is this phenomenon captured by the CG model?
  • The introduction needs to be expanded to include similar computational works on elastomers.  

Author Response

All the responses are included in the uploaded docx file.

Reviewer 2 Report

The authors developed a machine learning framework for predicting tensile stress value of natural rubber system, based on the data obtained from coarse-grained molecular dynamics simulations. The target for this research is to study/understand the relationship between the structural parameters of NR (Hydrogen bond strength, protein/phospholipids concentration and non-hydrogen bonding interaction strength) and corresponding tensile stress value (@600%). Due to the expensive computation for performing tensile test of the NR system, the authors performed series of data augmentation techniques to expand the data set for training the ML model and they found the optimal parameters for the XGB model and the predicting results are impressive. The authors certainly put a lot of effort in optimizing the data augmentation and the ML training algorithm for best fitting results. However, regarding the actual MD simulation for describing the NR mechanical behavior, the authors should put even more efforts since the quality of the MD simulation data determined the quality of the study. The whole study looks like a benchmark for showing the results of various data augmentation techniques and ML fitting without thinking much about target system to study. Therefore, I don’t recommend the paper published on Polymers in current form, unless the authors could address the following issues:

  1. In the coarse-grained MD model for describing the NR system, the authors chose three parameters (hydrogen bond interaction strength, protein/phospholipids concentrations and non-hydrogen bond interactions) as the target parameters for ML modeling. Is there any good reason for choosing those parameters? It is obvious that the increase in the hydrogen bonding strength between these polar groups/molecules in the system will lead to an improvement mechanical strength for same system, due to the stronger attraction pairs between them. Similar principle applies to the interaction strength between non-hydrogen bonding interaction as well. I am just feeling the target parameters for optimization could be too simple and why we need to apply Machine Learning (such a powerful tool) to describe such a simple system. In most cases, people chose ML to solve a problem when there are many input factors/parameters and many results to fit. The more complex for the system, the more benefits one can get from applying ML model. For describing/fitting such simple system, traditional multiple variable fitting method could do very well without such amount of effort. I could understand the authors want to emphasize the data processing flow and the ML framework for the potential more complex system, but what reader want to see is how well this framework do for solving complex system instead of using such powerful method to solve such a simple system. To give an intuitive example of what I mean a complex problem, here, I list some properties of the natural rubber system that fit the application of using the ML framework: (a) Chain length distribution effect, (b) crosslinker concentration, (c) protein group location/distributions, (d) non-linear deformation region, (e) Poisson ratio of the rubber network etc. Those properties are not easy to obtain/describe both in theoretically and experimentally, so it is meaningful for applying ML technique to describe them. Not only that, for a specific system with same chemical groups, but the strength of the hydrogen bonds interaction is also fixed, then why use this parameters as an target parameter for fitting the stress data. Therefore, choosing hydrogen bond strength and non-hydrogen bond strength as controlling parameters to model tensile stress does not make sense.
  2. In describing the coarse-grained model, the interactions parameters are shown in table 1 and 2. Please check the items in these tables, some interaction pairs are missing. Please show all pair interaction parameters (all combinations) used in the simulation, so the simulation could be reproducible.
  3. Please show the source or reason for how the interaction parameters (eps=3.8 and 0.38, sigma_0=4.89 A, shown in table 1 and table 2) are obtained and validated for this NR model. Does the equilibrium density or dynamics fit the finer model? Does the NR system in this coarse-grained model still have a Poisson ratio of 0.5? or the authors just assume the system has a Poisson ration of 0.5.
  4. I notice that the bonding interactions in the NR chains are modelled using harmonic bond potential. In the paper, the authors describe this as elastic bond, however, the reference for this part is from Kremer’s paper which is know for introducing FENE bond potential for describing the bond in modeling polymer chains. This is an inconsistency. I also noticed that the NR system was deformed to 600% when doing uniaxial deformation. Using harmonic bond potential could lead to a serious bond-crossing problem when the systems are in non-equilibrium state such as uniaxial deformation used in this paper. When polymer chain was stretched too much, harmonic bond potential cannot hold the bonding distance and lead to the polymer chain cross without breaking the bond. This will lead to a decreasing in the entanglement and significantly underestimate the tensile stress value. That is the reason why FENE bond potential [ref 24] is proposed and well adopted when performing polymer stretching simulations. Please recalculate all the tensile test simulations with the correct setup.
  5. The authors emphasized the MD simulation is very expensive, I cannot agree more. But please note that just because it is very expensive, we should do it properly to make it worth the cost. Beside all these issues mentioned above, the language in this paper is also a big problem. Please find a native/professional English speaker to go over the whole paper to polish the language.

Author Response

(The authors gave the same response as above.)

Round 2

Reviewer 2 Report

OK, I have three comments below for the revised manuscript,

(1) At the introduction part, please adding some words about the importance of the data augmentation for machine learning model and what could happen to the ML model training if the number of the training datasets are not enough. And give the logic reason you design your ML framework like that.

(2) I am not going to argue too much about the correct way to perform the MD simulation like that. I was hoping the authors use Kremer's model directly, so no one will argue about this simple model for polymer mechanical analysis. I can understand no model is perfect, but there are some important points that need special attention, such as the Poisson ratio calculation and the stress measurement from MD simulation. In most case, especially for polymer network, due to the randomness of the crosslinkers, the Poisson ratio of the network may not equal to 0.5 (especially for polymer with attractive interaction). So, when calculating the stress, if one assumes it equals to 0.5, the calculated stress could be wrong (or different from experimental measurement). So, I suggest the authors minimize the discussion about the MD calculation results (IMO, the MD simulation result is essentially a benchmark in this work) and pay more attention on describing the ML model framework and how to perform data augmentation of the MD simulation data.

(3) Language needs polishing

Author Response

(The authors gave the same response as above.)
